# Chronic Neutrophilic Leukemia: Advances in Diagnosis, Genetic Insights, and Management Strategies

**DOI:** 10.3390/cancers17020227

**Published:** 2025-01-12

**Authors:** Ismail Elbaz Younes, Pawel Mroz, Mehrnoosh Tashakori, Amira Hamed, Siddhartha Sen

**Affiliations:** Department of Laboratory Medicine and Pathology, University of Minnesota, Minneapolis, MN 55455, USA; pmroz@umn.edu (P.M.); tasha003@umn.edu (M.T.); hamed022@umn.edu (A.H.); sen00037@umn.edu (S.S.)

**Keywords:** chronic neutrophilic leukemia, myeloproliferative neoplasms, *CSF3R*, T618I, molecular pathogenesis, targeted therapy

## Abstract

Myeloproliferative neoplasms (MPNs) represent a subset of blood cancers, a member of which is chronic neutrophilic leukemia (CNL). This entity is very rare in comparison to the more common MPNs. The increased usage of next-generation sequencing (NGS) has increased our diagnostic and prognostic ability regarding this entity. There is no specified treatment approach for CNL; however, there are clinical trials that show promising results and experimental drugs that are being investigated to reach a unified treatment approach. This review aims to comprehensively review this rare entity in terms of epidemiology, diagnosis, pathogenesis, and treatment.

## 1. Introduction

MPNs include a group of bone marrow disorders with an underlying pathology based on the expansion of clonal hematopoietic cells leading to bone marrow hypercellularity, effective hematopoiesis, and an increase in number of one or more of the blood lineages with no apparent dysplasia. MPNs are subdivided into several different diseases based on morphological, cytogenetic, and molecular criteria.

One of the rarely encountered types of MPNs in routine hematopathology practice is CNL, with an estimated incidence of 0.1 cases per 1,000,000 people and a slight male predominance.

CNL was first described by Tuohy [1] in 1920, when he described a case presenting with massive splenomegaly and an overwhelming increase in mature polymorphonuclear neutrophils. Later, in 1932, a case report by Emile-Weil [2] in France described what is now thought to be a CNL and gave it the name of “La Leucémie Myélogène à Polynucléaires Neutrophiles”. However, it was not until 1964 that the term chronic neutrophilic leukemia was used when Tanzer et al. [3] reported a case with splenomegaly, increased neutrophils, and the absence of both the Philadelphia chromosome and bone marrow fibrosis.

According to Surveillance, Epidemiology, and End Results (SEER), the median age of diagnosis of CNL is 73 years old, which is slightly higher than that reported by Elliott et al. [4], who reported only 40 cases, whereas SEER reported 73 cases [5].In the National Cancer Database (NCDB), the median age was 70 years in 121 reported cases.

This rare entity has specific diagnostic criteria according to the fifth edition of the World Health Organization (WHO) Classification of Hematolymphoid Tumors and the International Consensus Classification (ICC) that combine morphologic, cytogenetic, and molecular findings [6,7].

According to ICC, there is a difference in the white blood cell (WBC) cutoff depending on the colony-stimulating factor 3 receptor (*CSF3R*) gene mutation status; in the presence of a *CSF3R* mutation, the WBC cutoff should be ≥13 × 10^9^/L, whereas in the absence of a *CSF3R* mutation, the WBC cutoff should be ≥25 × 10^9^/L. There is also another subclassification depending on the blast count; 10–19% blasts is called the accelerated phase and ≥20% blasts is called the blast phase.According to WHO, there is no accelerated or blast phase, and the WBC cutoff is ≥25 × 10^9^/L regardless of the *CSF3R* mutation status.

## 2. Disease Presentation

Most of the patients are asymptomatic and are incidentally found to have leukocytosis during a complete blood count (CBC). CNL patients might present with constitutional symptoms including fatigue, bone pain, pruritus, easy bruising, or gout. It can also be manifested as splenomegaly in two out of three patients or hepatosplenomegaly in 40% of patients [8]. Many studies have reported an association between CNL and bleeding disorders, which is considered one of the most common causes of mortality in this disease [9,10,11,12] and whose underlying mechanism may be exacerbated by chemotherapy [13]. Rare case reports showed CNL patients suffering from hemorrhagic events in the absence of both chemotherapy and thrombocytopenia, which could indicate that CNL can cause platelet dysfunction [14]. The mechanism of bleeding in CNL patients can be attributed to many mechanisms, including thrombocytopenia, platelet dysfunction, or the infiltration of tumor cells into the vessel walls. Additionally, Talon et al. suggested that enzymes secreted by the neoplastic cells might also play a role in the aforementioned bleeding disorders [15]. A subset of CNL patients, varying between 10 and 25%, undergo a leukemic transformation [16].

## 3. Diagnostic Findings

### 3.1. Laboratory Findings

Neutrophilia is considered the cornerstone of the diagnosis of CNL. Neutrophilia present in CNL is composed of mature neutrophils/bands (≥80% of the total WBCs, imperative criteria for diagnosis). In 2023, a case series study of 24 patients by Carreño-Tarragona et al. [8] revealed the following peripheral blood parameters (median) in CNL patients (see table below).
Hemoglobin (g/L)**7.9**Platelets (×10^9^/L)260WBCs (×10^9^/L)52.7Neutrophils (×10^9^/L)48Monocytes (×10^9^/L)1.0 (<10% of the WBCs)

Serum vitamin B12 has been reported to be high in CNL cases [17,18]. Transcobalamin I and III are reportedly high due to their release from mature granulocytes; however, this does not affect prognosis or treatment [19]. Serum granulocyte colony-stimulating factor (G-CSF) is found in low levels in CNL [20], whereas G-CSF is usually elevated in paraneoplastic leukemoid reaction [21]. In a study by Oogushi et al., neutrophil function in CNL was examined through several key tests, including chemotaxis, chemiluminescence, and nitroblue tetrazolium (NBT) dye reduction. While neutrophil function was normal, it was noted that leukotriene B4 production was decreased, and the significance of this finding is unknown [22]. Additionally, uric acid and lactate dehydrogenase (LDH) are usually elevated due to the high cell turnover. Leukocyte alkaline phosphatase score is elevated in CNL [23], in contrast to chronic myeloid leukemia (CML).

### 3.2. Morphological Findings

#### 3.2.1. Peripheral Blood

There is mature neutrophilia with band forms, which must account for more than 80% of the total white blood cell percentage count. Neutrophils might show abundant primary and secondary granules and Dohle bodies, similar to reactive changes. Toxic granulations and Dohle bodies are more common in plasma cell–associated leukemoid reactions rather than in CNL [24]. There is no dysplasia, no absolute basophilia, and no absolute eosinophilia. Absolute monocytosis can be present, although the number of monocytes should not exceed 10% of total white blood cells. Circulating immature granulocytes including promyelocytes, myelocytes, and metamyelocytes should not exceed 10% of the WBCs, and blasts are rarely found in peripheral blood. Red blood cells and platelet morphology are usually within normal limits.

#### 3.2.2. Bone Marrow

The bone marrow is usually hypercellular (>90%) with granulocytic predominance and a myeloid to erythroid ratio reaching 20:1 (See Figure 1A–D). In very rare occasions, megakaryocytic proliferation and neutrophilic phagocytosis by histiocytes are identified. Significant reticulin fibrosis or osteosclerosis is rarely reported [25].

## 4. Cytogenetic and Molecular Characteristics

CNL diagnosis requires the absence of t(9::22) (*BCR-ABL1*) translocation verified by cytogenetics or molecular studies. The majority of CNL cases havenormal karyotype. Cytogenetic studies may reveal additional chromosomal rearrangements including del (20q), del (11q), loss of chr.17, or trisomy of chromosomes 8, 9, or 21.

Molecular studies for the detection of pathogenic variants in the *CSF3R* gene can be performed by means of Sanger sequencing, pyrosequencing, high-resolution melting curve analysis, or most commonly via next-generation sequencing (NGS).

### 4.1. CSF3R

The *CSF3R* gene is composed of 17 exons and is located on chromosome 1p34.3. *CSF3R* gene encodes a receptor for G-CSF, which plays a prominent role in granulopoiesis. Several studies showed that mice deficient in G-CSF develop chronic neutropenia and impaired neutrophil immobilization [26] and that neutrophils deficient in G-CSF have increased apoptosis [27]. Wild-type *CSF3R* consists of three different domains (extracellular, transmembrane, and cytoplasmic), which are responsible for ligand binding, signaling activation, and termination of the receptor [28]. *CSF3R* is similar to other class I cytokine receptor family members in lacking intrinsic kinase activity; however, it activates *JAK-STAT*, MAPK/ERK, and PI3K/AKT signaling pathways through forming homodimers which bring the cytoplasmic domains closer together [29].

*CSF3R* mutations are found in many diseases other than CNL, including severe congenital neutropenia (SCN), which can play a driving role in the development of different myeloid neoplasms like acute myeloid leukemia [30]. A study by Olofsen et al. revealed that the truncated mutation d715 (Q739*) in *CSF3R* causes a pro-inflammatory state [31]. There is a subtype of SCN that is caused by the loss of function mutations in *CSF3R* and does not respond to G-CSF; while these patients showed peripheral neutropenia, the bone marrow did not show the maturation arrest that is seen in other SCN patients [32]. *CSF3R* mutations are also found in atypical chronic myeloid leukemia [33] (referred to as myelodysplastic/myeloproliferative neoplasms with neutrophilia in the fifth edition of WHO), chronic myelomonocytic leukemia (with CMML conferring poor prognosis) [34], hereditary neutrophilia (T640N, also known as T617N), acute myeloid leukemias, and myelodysplastic syndromes. A recent study showed that nine cases of AML had *CSF3R* mutations; however, eight of those nine cases had low VAFs, suggesting that they were subclones, while the other case had a VAF of 40%, and it had many other co-mutations such as *TP53* [35]. A study by Wang et al. showed a CNL patient with a compound CSF3R mutation and co-occurring T lymphoblastic leukemia [36]. The occurrence of lymphoma with CNL is a rare event; however, a study conducted in 2011 showed that the risk of development of lymphoid neoplasms in MPN patients was 2.79-fold higher [37]

There are two different types of mutations affecting the *CSF3R* gene:Point mutations usually affect the extracellular or transmembrane domains of the receptor, causing its activation regardless of the presence of the ligand as it activates the *JAK-STAT* pathway [38]. The most common mutation is p.T618I, and others, such as p.T615A and N610H, have been identified as well. Patients with these mutations can benefit from JAK-kinase inhibitors. While *CSF3R* mutations are the most common mutations in CNL, a *CSF3R* mutation (p.P733T) is more commonly found in CMML rather than in CNL [39].Frameshift or nonsense mutations lead to a premature stop codon, causing truncation of the cytoplasmic tail of the receptor [40] and leading to receptor overexpression and ligand hypersensitivity. This process occurs through the activation of tyrosine kinase nonreceptor 2 and SRC family kinases.

A recent study by Lance et al. reported the hereditary transmission of the T618I mutation in the *CSF3R* gene, which spanned four generations and several family members presented with CNL through increased expression of MCL1 (anti-apoptotic protein); however, there was no progression to acute leukemia [41].

### 4.2. ASXL1

Additional sex-combs like 1 *(ASXL1)* is the most commonly mutated gene after *CSF3R* in CNL. Most of ASXL1 mutations are truncating mutations and occur in the central intrinsically disordered region. In a recent study by Latacz et al., it was shown that one of the most common mechanisms for the pathogenesis of *ASXL1*-mutated neoplasms is increasing the H2A deubiquitination activity of BAP1 and boosting the expression of myeloid neoplastic genes through the formation of strong phase-separated condensates [42]. *ASXL1* mutations in myeloid neoplasms usually cause dysplasia [43,44]. When *ASXL1* is co-mutated with *CSF3R*, it causes increased myeloid differentiation rather than dysplasia. Different types of mutations were identified, including a truncation mutation in exon 12 [45], which was also reported by Abdulbaki et al. in a patient with CNL associated with thrombocytosis [46]. A truncation mutation in exon 13 was also reported by Elliott et al. [47]. Mutations in *ASXL1* and thrombocytopenia are considered to be adverse risk factors in CNL patients [47]. The prevalence of *ASXL1* varies among different studies between 40 and 80% [8,48,49]; however, Langabeer et al. reported the absence of *ASXL1* mutations in their study with a sample size of four patients only [50]. *ASXL1* plays a role in the increased production of malignant neutrophils in CNL patients with *CSF3R* mutations by repressing the production of MYC transcripts, which normally would lead to decreased myeloid differentiation [51]. *ASXL1* mutations are considered high-risk mutations in other MPNs that are *BCR-ABL1*-negative, such as polycythemia vera, essential thrombocythemia, or primary myelofibrosis [52]. A study by Awada et al. showed that the presence of *ASXL1* mutations in Philadelphia-negative MPNs is considered to be a risk factor for bleeding; however, in that study, no CNL patients were studied [53]. A large patient cohort composed of 1053 MPN patients showed that 27% of these patients harbor an *ASXL1* mutation. The most common mutation in this cohort was c.1934dupG (p.G646Wfs). They also found that there is a strong association between this mutation and *STAG2* mutations in MPN patients [54].

### 4.3. SETBP1

SET binding protein 1 *(SETBP1)* mutations occur more commonly in *CSF3R*-mutated CNL cases [55]. *SETBP1* mutations (G870S, which is the most common mutation in this gene, along with D868N, in myeloid neoplasms) play a synergistic effect with *CSF3R* in the pathogenesis of CNL [40,56]. Additionally, it was found by Zhang et al. [57] that patients with *SETBP1* mutations are found to have higher hemoglobin levels and platelet counts. The prevalence of *SETBP1* mutations in CNL is difficult to determine accurately, as reports vary significantly. Some studies estimate a prevalence as low as 14%, while others report rates as high as 56% [8,40,45,48,49]. This variation is likely due to the limited sample sizes in these studies. *SETBP1* mutations are more commonly seen in patients with T618I *CSF3R* mutation, and this is expected since T618I mutation is the most common mutation in CNL patients. A meta-analysis by Shou et al. reported that the presence of *SETBP1* has no impact on the prognosis of CNL patients [58]; however, a study by Elliot et al. revealed that *SETBP1* might play a role in leukemic transformation [47].

### 4.4. TET2

One of the most commonly mutated genes in clonal hematopoiesis of indeterminate potential (CHIP) [59] is commonly mutated in CNL, unlike DNA methyltransferase 3 alpha (*DNMT3A*) (the most common mutated gene in CHIP), which is less commonly mutated in CNL patients [40]. A study by Zhang et al. revealed that 20% of CNL cases have ten-eleven translocation-2 (*TET2)* mutations, while also showing that only 5.7% of patients had *DNMT3A* mutations [57], and another study by Meggendorfer et al. [48] revealed that 30% of cases carry *TET2* mutations. Since *ASXL1* and *TET2* mutations are common in CHIP, CNL patients carrying these mutations might acquire *CSF3R* mutations later, leading to the existing CHIP’s neutrophilic phenotype [40].

### 4.5. SRSF2 and U2AF1

Mutations in the spliceosome complex genes, such as serine/arginine-rich splicing factor 2 (*SRSF2),* have been detected in CNL patients at a frequency ranging between 20 and 40% [8,48].

Another gene of the spliceosome complex that plays a role in CNL is U2 small nuclear RNA auxiliary factor 1 (*U2AF1)*. A study by Dao et al. revealed that 4 out of 10 cases of *CSF3R* T618I-mutated myeloid neoplasms carried *U2AF1* mutations (these cases were not further characterized as to whether they were CNL or not in the original study) [60]. Another multi-institutional study, however, showed that the frequency of *U2AF1* mutations in CNL was ~13% [8].

### 4.6. RUNX1 and GATA2

Runt-related transcription factor 1 *(RUNX1)* and GATA binding protein 2 (*GATA2)* both play an important role in hematopoiesis, having been identified in CNL patients at 4.3% and 12.8%, respectively [8,57]. A study by Stoner et al. showed that three CNL patients developed disease progression after acquiring *RUNX1* mutations (164_G165insA, splice site, and R166fs*47 in one patient and S141L in another). One of these patients progressed to the blast phase (S141*), suggesting that acquiring *RUNX1* mutations might be an adverse risk factor and is indicative of disease progression [61]. A study by Yokota et al. shows that *RUNX1* mutations in MPNs are associated with blast transformation [62].

A recent case report by Nooruddin et al. showed that acquiring *GATA2* mutations and increased variable allele frequency (VAF) of *RUNX1* mutation might play a role in the pathological development of CNL and ruxolitinib resistance [63]. Another case report showed a CNL patient that progressed to the blast phase of B-lymphoid lineage following the development of a *RUNX1* mutation [64]. Another case reported a CNL patient with both a germline and a somatic *CSF3R* mutation (W791* and T618I) who developed mixed phenotypic acute leukemia upon acquiring a *RUNX1* mutation [65].

### 4.7. EZH2 and KDM6A

Enhancer of zeste 2 polycomb repressive complex 2 subunit *(EZH2)* and lysine-specific demethylase 6A (*KDM6A)* mutations were reported by Dao et al. with unclear significance [60]. A case report showed a patient with CNL and an *EZH2* mutation who received several targeted therapies that showed limited effectiveness. However, the reported patient also had *ASXL1* and *SETBP1* mutations, hence this poor drug efficacy might be related to the other mutations [66].

### 4.8. NRAS

NRAS proto-oncogene, GTPase *(NRAS)* mutations were also identified in CNL patients. These mutations can lead to activation of the RAS pathway, which would lead to the activation of the downstream mitogen-activated protein kinase (MAPK) signaling pathway, causing bone marrow proliferation [67]. A study by Zhang et al. revealed that the prevalence of *NRAS* mutations is around 9.5% [57], comparable to another study that showed that 8.7% of CNL patients in their cohort had *NRAS* mutations [8]. Both studies show that *NRAS* mutations have lower overall survival rates; however, the former study by Zhang et al. showed *NRAS* mutations were associated low hemoglobin levels, low platelet counts, a low neutrophil percentage, and a high monocyte percentage. *PTPN11* is also involved in the RAS pathway—mutations which are more common in juvenile myelomonocytic leukemia; however, it has been reported in cases of CNL [8,68,69].

### 4.9. CALR

Calreticulin *(CALR)* mutations are fairly rare in CNL. A study in 2014 by Lasho et al. reported a novel missense mutation (*CALR* E398D c1194G > T) that is different from the pattern seen in different MPNs that have either insertion or deletions resulting in frameshift mutations, which may result in a different pathogenic pathway than the norm seen in frameshift deletions in *CALR* [70]. Another study cohort shows the same *CALR* mutation in one of their CNL patients [47]; however, a study by Cue et al. reported the presence of a frameshift mutation in *CALR* (c.1154-1155insTTGTC) [71]. The importance of either mutation in *CALR* regarding diagnosis or prognosis is unclear.

## 5. Molecular Pathogenesis

The pathogenesis of CNL is dependent on different types of mutations as discussed above; however, the evolution of the clones in CNL and their chronological pattern is important.

A study by Zhang et al. [57] showed that mutations of *EZH2*, *SETBP1*, *TET2*, *U2AF1*, and *SF3B1* usually have VAFs that do not cross 50%, suggesting that these mutations were acquired by an earlier clone. On the other hand, *ASXL1*, *CSF3R*, *NRAS*, *SRSF2*, and *CBL* mutations had a higher variety of VAFs, suggesting that they might represent the earlier founder dominant clone in some cases or a subclone in other cases. Nooruddin et al. reported a patient that had a *RUNX1* mutation at a low VAF, afterwards the VAF increased when the patient developed blast transformation, also the patientacquired other mutations in *KIT* and *GATA2*, which suggests that the acquisition of these mutations might play a role in leukemic transformation [63]. Langbeer et al. showed that an increased VAF of the *CSF3R* mutation (T618I) and the gaining of *NRAS* mutations are associated with leukemic transformation, and interestingly found that the loss of *CBL* mutations is associated with clonal evolution [50].

Clonal evolution of CNL is not only dependent on the acquisition of mutations but also on the gain of cytogenetic abnormalities associated with blast transformation. A study found two cases, one with the addition of chromosome 19 and another with the addition of (12) (p11.2), that were detected at the time of blast transformation but not at the time of diagnosis [44]. A study by Elliot et al. showed that trisomy 21, del12p, and monosomy 7 were acquired during the disease course of three different patients [4]. Another study showed a patient who developed blast transformation with the acquisition of monosomy 5 and monosomy 7. CNL patients with disease progression tend to develop blast transformation; however, in certain cases, they might develop other myeloid neoplasms such as CMML [47]. CNL can also arise from different myeloid neoplasms; a recent case report showed a patient who was diagnosed with MDS with *U2AF1* and *SETBP1* mutations who, 3 years later, developed CNL and acquired *CSF3R* and *ASXL1* mutations [72].

### Pathogenesis

Multiple models are hypothesized to cause clonal evolution. Maxson et al. categorize these into three main models:

**Model A:** *CSF3R* mutations occur on top of a CHIP mutation, which is usually seen in genes that regulate epigenetic or splicing processes.



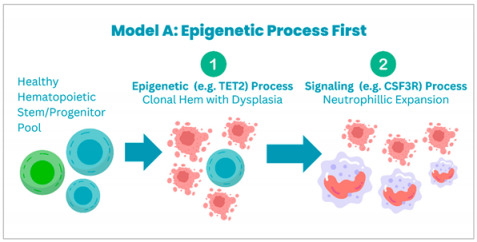



**Model B:** CSF3R mutations are considered to be the primary process, and later on, the acquisition of other mutations leads to clonal evolution.



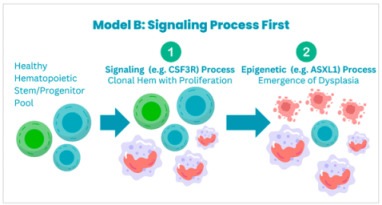



**Model C:** The third model postulates that distinct population of clonal cells acquire differnt mutations that lead to disease development; however, this seems less likely because the VAFs of the different mutations such as *CSF3R*, *ASXL1*, *SETBP1* have been reported to be similar [40].



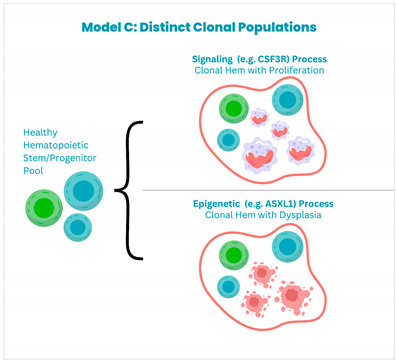



Another study by Fleischman et al. showed that the *CSF3R* mutation T618I is enough to cause lethal myeloproliferative disorder in a murine model [73].

## 6. Diagnosis Criteria per ICC and WHO 5th Edition the Diagnostic Criteria for CNL Is Summarized in the Table Below

### Differential Diagnosis

Reactive causes of neutrophilia include infection (bacterial or viral), inflammatory disorders, and drug effects such as corticosteroids, and hence a careful and comprehensive clinical evaluation is needed before rendering a diagnosis of CNL. An important aspect that needs to be ruled out is the G-CSF effect whether it is exogenous or endogenous. The bone marrow will usually show increased neutrophils with toxic granulation, monocytosis, and a left-shifted maturation of neutrophils. Some cases will show circulating blasts in the peripheral blood, and these neutrophils will be non-clonal, hence establishing that clonality might be helpful in certain cases or when the clinical history is unavailable. Leukemoid reactions which can occur due to different underlying diseases or conditions should be ruled out.
**Criteria****WHO 5th Edition** [7]**ICC** [6]LeukocytosisPresent, must be ≥25 × 10^9^/L, regardless of mutations≥13 × 10^9^/L for CSF3R- mutated cases, ≥25 × 10^9^/L in non-CSF3R mutated casesMarrow cellularityHypercellular, neutrophils + banded neutrophils constitute ≥80% of the white blood cellsNeutrophil precursors<10% of the WBCsNeutrophiliaPersistent for 3 months or more, splenomegaly with no identification of reactive causes in mutation negative casesDysplasiaNot presentMonocytes<10% of peripheral blood monocytesRule outChronic Myeloid Leukemia (CML), polycythaemia vera, essential thrombocythaemia, or primary myelofibrosis, myeloid/lymphoid neoplasms with eosinophilia and tyrosine kinase gene fusions and exclusion of reactive neutrophilia.

Chronic myeloid leukemia (CML) usually presents with neutrophilia; however, in bone marrow biopsy, there will be a left-shifted leukocytosis with myelocyte bulge as well as small hypo-lobated megakaryocytes and basophilia. Additionally, genetic studies will reveal a Philadelphia chromosome t(9;22)q34.1;q11.2. It can be even more challenging in cases of the neutrophilic variant of CML, as it has robust neutrophilic differentiation, and hence it is important to test for all three breakpoints of the Philadelphia chromosome (p190, p210, p230). A significant amount of CNL cases present with mild to moderate anemia [8,74]. Platelet counts are usually normal; however, thrombocytopenia can be present in some cases, especially in the later stages and splenomegaly.

CMML, according to the new WHO edition and the ICC, absolute monocyte count decreased from ≥1 × 10^9^/L to ≥0.5 × 10^9^/L (if clonality and dysplasia are established), and hence it is more challenging to diagnose. However, monocytes must account for more than 10% of the total leukocytes, which is not the case in CNL. In addition, CMML shows dysplastic features in neutrophils and other lineages as well.

Atypical chronic myeloid leukemia is a subtype of MDS/MPN neoplasms, considered as a diagnosis of exclusion. This entity presents with leukocytosis and neutrophilia without significant monocytosis or basophilia or eosinophilia The significant neutrophilia is what might cause the confusion with CNL; however, the neutrophilia in this entity is left-shifted (<20% blasts), which should comprise more than >10% of the WBC count. The presence of significant dysplasia in the granulocytic lineage (erythroid and megakaryocytic lineages might be dysplastic as well) is important, as CNL lacks dysplasia. Since aCML is a subtype of MDS/MPNs, cytopenias and cytosis are to be expected.. *SETBP1* and *ETNK1* mutations are more common in aCML versus CNL, even though there are cases of aCML with *CSF3R* T618I mutations [75]; the presence of this mutation should make the pathologist pause and consider a CNL diagnosis. Even though CSF3R T618I mutation is more common in CNL, there is a recent study that showed the presence of this mutation in five out of six patients with aCML and four out of six patients with CNL [33]. A multicenter study led by Carreño-Tarragona et al. [8] suggested that both entities should be considered as one based on similar clinical and molecular profiles; however, Trembaly et al. argued in opposition to this due to the different overall survival [76]. This is also supported by another study by Sun et al. that revealed that each entity has a distinct clinical profile, with lower hemoglobin and platelet levels in aCML in comparison to CNL. They also showed that these entities have different molecular and cytogenetic profiles, with abnormal karyotypes being more commonly found in aCML [77].

*BCR-ABL1*-negative MPNs:○Polycythemia vera (PV) usually presents with a hypercellular marrow which also might show panmyelosis; however, the increased hemoglobin levels coupled with the morphology of the megakaryocytes are important differences coupled with the presence of a *JAK2* mutation which is more common in PV.○Pre-fibrotic primary myelofibrosis (PMF) usually presents with a hypercellular marrow. One of the minor criteria is leukocytosis; the megakaryocytes are clustered and have an atypical morphology, while on the other hand, the megakaryocytes in CNL are usually normal.

Plasma cell neoplasms can present with neutrophilia, which might cause a diagnostic dilemma. Immunohistochemical stains and flow cytometry can be performed to exclude an underlying plasma cell neoplasm, as its presence will generally exclude a diagnosis of CNL unless the clonality for CNL can be documented independent of the underlying plasma cell process. Some case reports have shown the coexistence of plasma cell myeloma and CNL [78,79,80,81].

## 7. Prognosis

CNL has a poor prognosis and its overall survival is short. A population-based study of CNL in the United States showed that according to the SEER program and the NCDB, the overall survival rate is 1.8 and 2.2 years, respectively [5]. Another review in 2002 showed that 5-year-old survival was 28% [82]. A study by Elliot et al. revealed that blast transformation occurs in many patients, with a median of 21 months from the time of diagnosis [4].

Many important factors play a role in prognosis, such as the type of mutation of *CSF3R.* It was found that patients harboring *CSF3R* (T618I) mutations tend to have a worse prognosis than patients harboring other *CSF3R* mutations [49]. *ASXL1* mutations also tend to have a worse prognosis [47]. There are conflicting studies regarding *SETBP1* mutational status, with some studies confirming that it carries a worse prognosis [40], while another study showed that there is no prognostic disadvantage in CNL patients with *SETBP1* mutations [58].

Leukocytosis was also found to have a detrimental effect on prognosis. Patients with >50 × 10^9^/L had a median overall survival of 11 months vs. 39 months for patients with <50 × 10^9^/L [83]. Elliot et al. also found that thrombocytopenia is considered a poor prognostic factor [47]. A study by Zhang et al. found that patients with *NRAS*, *ASXL1*, *GATA2*, and *DNMT3A* mutations had a shorter overall survival; on the other hand, *CBL* mutations tend to be associated with a more favorable survival outcome [57]. A recent score by Szuber et al. was devised to subdivide CNL into different groups according to prognosis:Thrombocytopenia (platelet count of <160 × 10^9^/L), 2 pointsLeukocytosis >60 × 10^9^/L,1 point*ASXL1* mutation status, 1 point

If patients had zero to one point, they were considered low-risk, while those with two to four points were considered high-risk. Patients who were considered high-risk are more closely monitored for disease progression and they might be given a stem cell transplant earlier, since they are already at a higher risk [49].

## 8. Treatment

There is currently no standard of care for CNL due to the small number of cases, limited survival rates, and dispersion of patients. The only therapy designed to provide a cure is hematopoietic stem cell transplant.

### 8.1. Traditional Therapy

Splenic irradiation and splenectomy of CNL patients has been used since 1979 as a palliative measure in patients with symptomatic splenomegaly [84]; however, due to post-operative neutrophilia, splenectomy is no longer recommended for CNL patients [85]. Splenic irradiation might still be considered in CNL patients with symptomatic splenomegaly that does not respond to other therapies [86]. Another approach is using oral cytoreductive chemotherapy such as hydroxyurea, which achieves its goal by reducing leukocytosis and splenomegaly. This approach has been used for a long time; however, the drawback is that it has a transient effect and is not long-lasting (median of 12 months). There is a subset of patients that are refractory to this treatment from the outset of the disease [4]. Another study in 2019 revealed that hydroxyurea was the initial therapy for CNL patients even though the effect is transient, leading to the use of other lines of treatment [87]. Interferon alpha has also been used for CNL; it is a drug that has shown remission in CNL patients. Notably, this observation has only been published in case reports [12,88,89,90]. Other forms of interferons such as peginterferon alfa-2a (Pegasys) and ropeginterferon alfa-2b (ropegIFN; Besremi) are used in other MPNs such as PV; however, there are no reports on its usage in CNL [91]. It would be interesting to see if these drugs actually have an effect in CNL patients, as interferon alpha has shown good results in CNL.

Standard induction 7 + 3 chemotherapy induction has been used in the blast phase in CNL. Hasle et al. reported a blast-phase CNL case that reverted to the chronic phase through the standard induction regimen [85]. This finding, however, was found to be inconsistent with a study by Elliot et al., which reported that the patients were refractory to such an approach [4]. A review by Tefferi et al. showed that blast-phase MPNs should receive ventoclax plus hypomethylating agents regardless of being transplant-eligible [92]. This study, however, focused on PMF, essential thrombocythemia, PV, and MPN-unclassifiable cases.

### 8.2. JAK Inhibitors

Most cases of CNL are directed by *CSF3R* mutations which activate the *JAK-STAT* pathway; hence, it is postulated that JAK inhibitors can work on CNL patients. Even though ruxolitinib (*JAK1/2* inhibitor) is not FDA-approved for CNL cases, it has been used in murine studies and CNL patients harboring a *CSF3R* T618I mutation [38,93]. Another study showed that it also worked in a patient harboring compound *CSF3R* mutations (T618I and Q749X) [94].

In a phase II trial of ruxolitinib for CNL involving 21 patients, the overall response rate (ORR) was 58%. This included four patients achieving complete remission (CR) and nine achieving partial remission (PR). Response rates were markedly lower in patients with wild-type *CSF3R* (8%). Median overall survival (OS) was 18.8 months, with responders showing a longer median OS of 23.1 months compared to 15.6 months for non-responders. Ruxolitinib reduced the *CSF3R* T618I allelic burden [87]. Fleischman et al. demonstrated the effect of ruxolitinib in murine models harboring CSF3R T618I (lower WBC count and reduced spleen weight) [73]. Gunawan et al. reported a case with compound *CSF3R* mutations that received ruxolitinib in which the patient had an excellent hematologic response and decreased VAFs of both *CSF3R* mutations from 50% to 8%; however, after 9 months, the patient stopped responding to the drug and relapsed [95]. Koppikar et al. speculate that the mechanism of resistance to ruxolitinib might be due to heterodimeric JAK-STAT activation [96]. Ruxolitinib does not just target the T618I mutation in *CSF3R* but also can target other mutations such as p.N579Y [97]. Preliminary data from a phase II clinical trial of fedratinib (a *JAK2* inhibitor), which is approved for higher-risk myelofibrosis (NCT05177211), showed promising clinical efficacy in CNL. It also has advantages over ruxolitinib, as it possesses a more comprehensive kinase inhibition profile and effectively inhibits *FLT3* and *BRD4*. Moreover, it suppresses c-MYC expression (51% average decrease in c-MYC in all cases in the clinical study) [98]. A case report showed that a CNL patient with compound *CSF3R* mutations preceded by *DNMT3A* achieved hematologic remission following treatment with ruxolitinib (JAK1/2 inhibitor) [94]. Another study showed a different outcome for a murine model harboring compound *CSF3R* mutations, as all the mice treated with ruxolitinib succumbed to death within 7–8 weeks, which might suggest that it is not effective in CNL patients harboring compound *CSF3R* mutations [99]. There are variable responses to ruxolitinib in different CNL patients; however, the drawback is that most of these studies have a small number of patients. It has been reported that the concurrent *SETBP1* with *CSF3R* confers resistance to ruxolitinib [100]. Signal transducer and activator of transcription 3 (*STAT3*) mutations would cause a mechanism of resistance to ruxolitinib, as their mechanism of action is downstream of JAK-kinases, which avoids ruxolitinib effects. Other mutations have been postulated to occur in patients receiving ruxolitinib during disease progression, such as *STAG2*. *RUNX1* mutations have been detected during disease progression while the patient was receiving ruxolitinib; however, in the same study, two patients had *RUNX1* mutations and did not progress to AML [61]. Another study has shown that loss of ruxolitinib response was associated with increased frequency of *RUNX1* mutations and the development of *GATA2* and *KIT* mutations [63].

### 8.3. Novel Therapies

Dasatinib, an *SRC* kinase inhibitor, showed in vitro sensitivity to CNL lines with truncation mutations in *CSF3R*, in contrast to proximal membrane mutations in *CSF3R* [38]. Dasatinib is highly sensitive to *CSF3R* frameshift mutations [38,101]; however, it has been shown that in compound mutations (proximal membrane and truncation), this effect is nullified [94]. A case report showed that a patient with B-cell acute lymphoblastic leukemia harboring three truncation mutations in *CSF3R* achieved complete remission with minimal residual disease after chemotherapy and dasatinib [101].

Trametinib (*MEK1/2* inhibition) targets myeloid neoplasms harboring *NRAS* mutations, representing a subset of CNL patients [102]. Another study showed that trametinib targets either proximal or compound *CSF3R* mutations in murine models, which could indicate it is a potential therapeutic agent [99]. A recent study by Parducci et al. demonstrated the antineoplastic effect of aurora kinases in Ba/F3 expressing the *CSF3R* T618I mutation through a decrease in proliferative capacity and cell viability. In the same study, reversine (dual inhibitor for aurora kinase A and B) was found to induce *PARP1* cleavage, γH2AX expression, and decreased *STAT5* phosphorylation [103].

### 8.4. Stem Cell Transplant

Stem cell transplant is the onlytherapeutic option with curative potenital for CNL; however, 2 major concerns arise the complications that might arise with transplant such as graft versus host disease, graft failure and transplantation related mortalityThe other concern is the lack of recognition of this entity and the need to reach this diagnosis quickly, as it has a bad prognosis, which prompts accurate diagnosis and fast intervention. There are some case reports showing patients achieving complete remission following allogeneic stem cell transplant, showing the efficacy of this approach [85,104]. In a case series of 19 CNL patients, 2 patients had blast transformation and underwent stem cell transplant. One patient remained disease-free for 40 months; however, the other had venoocclusive disease and died of infection [49]. A review of stem cell transplants in CNL by Menezes et al. revealed that patients who received transplants during the chronic phase show significantly better outcomes, with 71% experiencing ongoing remission for over seven months [105]. A population-based study by Ruan et al. revealed that 2% of CNL patients (121 total patients) underwent stem cell transplants, with a 100% 5-year survival rate in all of them [5]. A similar retrospective study in Japan showed that between 2003 and 2014, five patients underwent a stem cell transplant. Two of the five patients achieved complete remission, one patient died of bleeding before achieving neutrophil engraftment, another patient died of sinus obstruction syndrome, and the last one showed no response to transplant and died of the underlying disease [106].

Given the abysmal prognosis of this disease, it is imperative to conduct more studies that focus on this disease, as there is no standard of care for this disease. The limitation that we might face in these clinical trials is the lack of cases given the rarity of this entity.

## 9. Conclusions

CNL is a rare myeloid neoplasm that has been discovered for over a century; however, due to the scarcity of cases and the divergent clinical picture, as well as the change in the criteria of diagnosis, it is hard to reach this diagnosis. The understanding of CNL pathogenesis is clearer now due to the advancements in molecular studies. There is currently no standard of care for CNL patients, and many drugs are in the experimental phase; however, reaching complete remission can currently only be achieved by stem cell transplant. The prognosis of this entity is abysmal, with many acquired molecular alterations causing disease progression. There is a significant need for improved treatment options for CNL. Current challenges include a limited understanding of the disease’s genetic complexities, which hinders the development of effective therapies. To address this, more studies are needed that specifically focus on CNL patients. Such research could uncover new genetic targets, paving the way for the creation of more effective treatments.

## Figures and Tables

**Figure 1 cancers-17-00227-f001:**
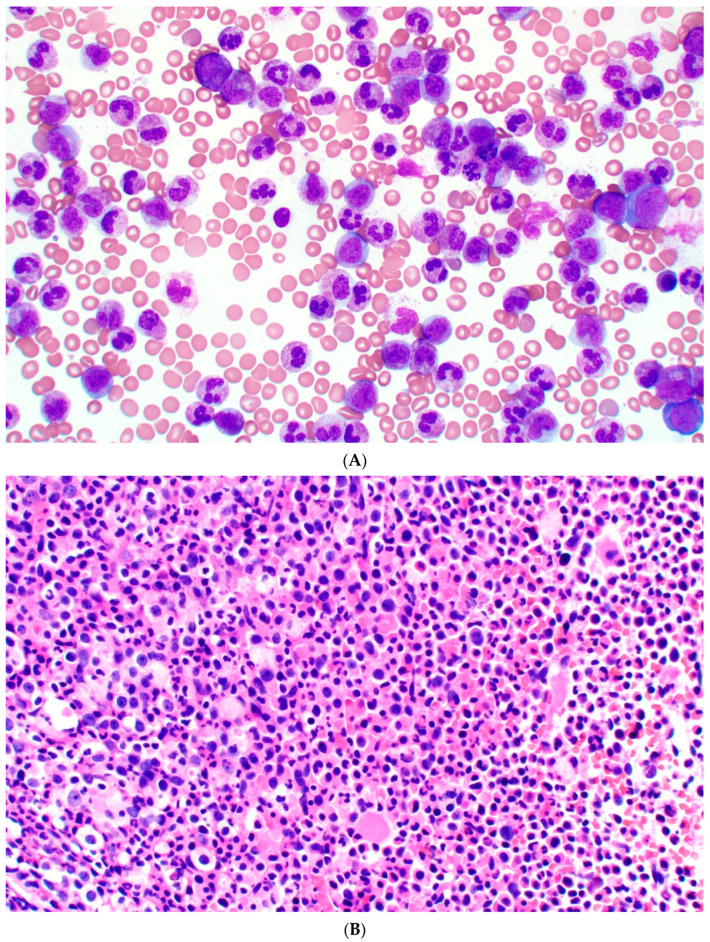
(**A**) High-power aspirate (500× Wright stain) showing a predominance of neutrophils and bands. (**B**) High-power bone marrow core biopsy (400× hematoxylin and eosin) showing trilineage hematopoiesis with predominance of neutrophils. (**C**) Low-power aspirate (200× Wright stain) showing a predominance of neutrophils and bands. (**D**) Low-power bone marrow core biopsy (40× hematoxylin and eosin) showing a hypercellular marrow.

## Data Availability

No new data were created or analyzed in this study. Data sharing is not applicable to this article.

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
