# Peer review of "Chronic Neutrophilic Leukemia: Advances in Diagnosis, Genetic Insights, and Management Strategies"

_cancers, 2025, doi:10.3390/cancers17020227_

Round 1

Reviewer 1 Report

Comments and Suggestions for Authors

Dr. Sen S and colleagues provide a clear and concise overview of chronic neutrophilic leukemia (CNL), describing it as a rare and distinct subset of myeloproliferative neoplasms (MPNs). The review effectively delves into the pathophysiology, clinical manifestations, and, most importantly, the role of genetic mutations in both diagnosis and prognosis. By focusing on the key genetic markers, the authors offer valuable insights into how these mutations influence disease progression and outcomes. Additionally, the review addresses the limited understanding of both short-term and long-term treatment strategies for CNL. Overall, this article is an essential read, well-organized, and easily accessible, providing important information for both clinicians and researchers in the field.

Author Response

Thanks a lot for your wonderful review. If there are any comments or edits we would be really happy to address

Reviewer 2 Report

Comments and Suggestions for Authors

Thank you for allowing me to review this paper some parts read eerily similar to "Szuber, Natasha, Michelle Elliott, and Ayalew Tefferi. "Chronic neutrophilic leukemia: 2022 update on diagnosis, genomic landscape, prognosis, and management." American journal of hematology 97.4 (2022): 491-505."

I would take precautions to ensure there are distinct differences in the manuscripts.

8.1 Regarding traditional therapy the first sentence is hard to understand. It notes splenic irradiation being used as a therapy and then notes that due to post-operative neutrophilia splenectomy is not recommended. I would assume it is known that splenic irradiation and splenectomy are two different modalities with different complications. Splenic irradiation is a less utilized modality due to lack of durable response and profound cytopenias. Splenectomy is not used due to morbidity/mortality as high as 30%. These needs to be clarified with references.

Pegasys/besremi  would argue is not a traditional therapy as its not used readily by clinicians. We do know in other MPNs this is one of the few disease modifying drugs and should be possibly highlighted separately. I would note with the adoption of besremi in PV, MPN and ET we need more studies utilizing this therapy.

8.3 Other therapies

No note of HMA or other therapies utilize in accelerated phase/blase phase. I think this is important if you're planning to transition to allohct.

8.4

I'm not sure those (limited numbers is real) are the real concerns for transplant. I think we know its a curative therapy but associated complications related to TRM, aGVHD and cGVHD are what give transplanters pause to proceed forward with transplanting CNL patients with a lack of data. The median age of 70 at diagnosis for CNL will not make these patietns inelgibile for transplant in high volume transplant centers utilizing RIC regimens. I think there is a lack of referral and recognition of CNL as an entity that requires urgent therapy due to poor prognosis. We need more studies in this space to define not if we should transplant these patient but how and when. I think acknowlegement in this section of this issue would be ideal.

I do think there should be a section for novel therapies and any noted trials if available. 

Author Response

We do appreciate your comments.

Regarding your first comment, we have separated splenic irradiation and splenectomy to remove any confusion.

Pegasys/besremi: We have added a part about those 2 drugs, however, there are no studies/clinical trials that test the usage of 1 or 2 of those drugs in CNL. We have addressed the fact that it would be a good idea to have such a study

HMA for blast phase: There have not been any reports to our knowledge about the usage of HMA in CNL, however,  we added a part addressing the usage of standard chemotherapy 7+3 in CNL patients

Stem cell transplant: We appreciate your comment about ineligibility for stem cell transplant and we have modified it accordingly

Novel therapy: There is a section for novel therapy but it was under other therapy so we changed it to Novel therapy. 

We also added a paragraph before the conclusion addressing the need to have more studies for this disease

Reviewer 3 Report

Comments and Suggestions for Authors

Dear authors,

  The aim of this review was to provide a comprehensive review of CNL, focusing on breakthroughs in genetic profiling, including novel mutations with potential prognostic value and implications for targeted therapy.

  The paper lacks critical review, with minimal practical value and the originality is quite low since reviews on the same topic were recently published.

  I regret to inform you that I don’t consider the manuscript acceptable for publication in its present form.

Author Response

Thanks a lot for your review. We regret that you believe that the paper lacks any scientific significance. We have addressed almost every aspect of CNL, from diagnosis to treatment. If you have any point in specific we would be open to edits and implementing them. Thanks again for taking the time to review the article

Reviewer 4 Report

Comments and Suggestions for Authors

The authors aim to summarize the available data on diagnosis, prognosis and management of this very rare disease. It would certainly make sense to add “prognosis” into the title.  

It is an important and relevant topic due to the rare and difficult-to-treat disease, but it is not original to the field as it does not really address a specific gap in this field and due to just currently published similar reviews to this disease (e.g. Szuber N. et al., Am J Hematol, 2024).

But the colleagues summarize in a very clear way the advances in the diagnosis of CNL especially with regard to the molecular genetic informations (NGS) which are available in the current literature. The authors also delineate accurately the prognostic relevance of the genetic findings although it is not emphasized enough that especially mutations like ASXL1, RUNX1 and DNMT3A are not a specific finding for CNL.

It is questionable if this work adds anything to the published material. But it gives a comprehensive Integration of the most important and up to now available data.

The differences of CNL and aCML  (MD/MPN-N) could be better elaborated as in both diseases similar genetic changes might occur.

The data they refer to is mostly gained via Case reports as well as in trials with very small numbers of patients. Some data is also only won by mouse models which is often not transferrable to humans. With regard to the management and treatment of the disease large-scale clinical trials are needed to validate the therapeutic potential of treatment options like JAK-Inhibitors which is addressed by the authors in their summary.

Reference list: Reference 1-3 is double; Reference 42 – only a very short abstract, its relevance should be questioned, maybe omitted.

 Comments on the text:

-       Line 7: please include the abbreviation MPN [Myeloproliferative Neoplasms (MPN) …]

-       Line 8: This entity is VERY rare …

-       Line 39-40: … I would put “La Leukemie myelogene a polynuclearies neutrophils” in brackets.

-       Line 47: omit “old”

-       Line 64: replace “picture” by “count”; I would also omit “many” 

-       Line 72: please include the reference after “…dysfunction.”

-       Line 85/86: high Transcobolamin levels are often increased in malignant diseases (especially solid tumors) – it would be interesting to explain the relevance for daily routine – is there a prognostic relevance in CNL (or if not), otherwise its just an enumeration of lab results not specific to CNL.

-       Line 91: …”it was noted that leukotriene B4 production was decreased” – why important/relevance?

-       Chapter 3.2.1: please add the literature (WHO/ICC).

-       Figure 1A/1B/1C/1D: please add stain and microscope

-       Line 131: I would add “The” in front of CSF3R gene and “on” chromosome 1p34.3 instead of “in”

-       Line 133: … “mice deficient”

-       Line 153: add “the” in front of CSF3R gene

-       Figure 2: I would try to find a better graphic

-       Line 307: The table should be improved – its not clearly arranged.

-       Line 326/327: “Platelet counts are usually normal; however, thrombocytopenia can be present in many cases, especially in later stages and splenomegaly.” à ??? inconsistent!

-       Line 377: “detrimental”, not “decremental”!

-       Line 384: in most countries thrombopenic means platelets <150 G/l, à <160???

-       Line 395: for such a procedure; rest of the sentence à better English. ???

-       Line 465: AND? Effect/consequence?

-       Line 481: I would change “a” in “one”

Additional Comments:

     The manuscript is generally well-written. However, proofreading is needed to correct grammatical errors, typos, and formatting issues.

Recommendation: I recommend that the authors revise the manuscript by addressing the points outlined above. Upon satisfactory revision, the manuscript might be suitable for publication.

Author Response

Thanks a lot for your extensive review.

ASXL1: We have added more data and did more research on this specific gene since it is the most common gene after CSF3R

RUNX1: We have added another study that shows that RUNX1 mutation is associated with blast transformation in MPNs

DNMT3A: It is rarely present in CNL so we mentioned that DNMT3A mutations might be present in 5% of patients, however, they do not have a prognostic significance

We have made all the grammatical changes that you have suggested. 

As far as the table we have added a small line in the first part of it to avoid the confusion, the rest of the table is actually the same in both ICC and WHO.

Figure 2 is actually not very important as it shows the IGV for this mutation and we figured it will not be very important to the reader so we removed the figure

We also delved more into the additional pathogenic mutations especially ASXL1.

We also elaborated on more points in order to make the distinction between aCML and CNL.

We do agree that most of the treatment regimens that we mentioned stem from case reports, however, this is mainly due to the rarity of the entity.